# Sampling on Metric Graphs

## Abstract

Metric graphs are structures obtained by associating edges in a standard graph with segments of the real line and gluing these segments at the vertices of the graph. The resulting structure has a natural metric that allows for the study of differential operators and stochastic processes on the graph. Brownian motions in these domains have been extensively studied theoretically using their generators. However, less work has been done on practical algorithms for simulating these processes. We introduce the first algorithm for simulating Brownian motions on metric graphs through a timestep splitting Euler-Maruyama-based discretization of their corresponding stochastic differential equation. By applying this scheme to Langevin diffusions on metric graphs, we also obtain the first algorithm for sampling on metric graphs. We provide theoretical guarantees on the number of timestep splittings required for the algorithm to converge to the underlying stochastic process. We also show that the exit probabilities of the simulated particle converge to the vertex-edge jump probabilities of the underlying stochastic differential equation as the timestep goes to zero. Finally, since this method is highly parallelizable, we provide fast, memory-aware implementations of our algorithm in the form of custom CUDA kernels that are up to ∼8000x faster than a GPU implementation using PyTorch on simple star metric graphs. Beyond simple star graphs, we benchmark our algorithm on a real cortical vascular network extracted from a DuMuX tissue-perfusion model for tracer transport. Our algorithm is able to run stable simulations with timesteps significantly larger than the stable limit of the finite volume method used in DuMuX while also achieving speedups of up to ∼1500x.

## 1 Introduction

Metric graphs, also known as quantum graphs (Kuchment, 2004), are geometric structures formed by gluing together one-dimensional segments of the real line at the vertices of an underlying graph, inheriting both the combinatorial topology of a graph and the smooth geometry of a real line. These objects have emerged as powerful tools for modeling complex systems in diverse fields, including physics, biology, and network theory. For instance, they are used to model nanoscale materials like carbon nanostructures (Amovilli et al., 2004), vascular networks (Carlson, 2006; Blinder et al., 2013), nerve impulse transmission (Nicaise, 1985), acoustics (Cacciapuoti et al., 2006), and traffic flow on road networks (Garavello & Piccoli, 2006). Specific applications for vascular networks include solving diffusion PDEs like the Fokker-Planck equation to simulate blood flow dynamics, drug delivery, or nutrient transport in the brain. For example, we run numerical experiments on the mouse cortical microvascular network studied in (Blinder et al., 2013) with boundary/pressure data estimates obtained from (Schmid et al., 2017; Schmid, 2017). In the case of road networks, applications include traffic flow simulations which involve solving conservation law PDEs. A common theme that we explore in this work is that these applications involve numerically solving a diffusion PDE, which can be done stochastically using sampling methods. We refer the reader to (Kuchment, 2002) for a comprehensive survey of the applications of quantum graphs. From a theoretical standpoint, the underlying metric structure of metric graphs allows for the analysis of differential operators (Mugnolo, 2014; Erbar et al., 2022) and stochastic processes (Freidlin & Sheu, 2000), enabling the study of phenomena such as diffusion, wave propagation, and random motion on networks.

Brownian motions on metric graphs, a canonical example of such stochastic processes, have been extensively studied theoretically through their infinitesimal generators (Kostrykin et al., 2007; 2010; Kostrykin & Schrader, 2006; Aleandri et al., 2020). However, practical algorithms for simulating these processes – essential for numerical studies and real-world applications – have remained underdeveloped. This gap is particularly consequential in modern computational statistics and machine learning, where efficient sampling methods on complex geometries are indispensable (Byrne & Girolami, 2013; Betancourt, 2017). For example, Langevin diffusions (Roberts & Tweedie, 1996), a class of stochastic differential equations (SDEs) central to sampling from high-dimensional distributions, have seen widespread adoption in Bayesian inference (Girolami & Calderhead, 2011) and molecular dynamics (Leimkuhler & Matthews, 2015). Extending these methods to metric graphs could unlock new applications in networked systems, such as diffusive transport in dendritic networks in neuroscience (Bressloff, 2014).

Despite progress in understanding the theory of SDEs on metric graphs (Freidlin & Sheu, 2000) – including vertex transition rules, Feller properties, and large deviation asymptotics – the numerical simulation of these processes has been largely unexplored. Existing numerical work on metric graphs has focused primarily on solving partial differential equations using finite element methods (Kravitz, 2022). Some of these methods, such as finite volume schemes, struggle to stably scale to finer meshes without requiring prohibitively smaller timesteps (LeVeque, 2002) and are also less amenable to parallelization on modern hardware (GPUs) compared to Monte Carlo methods. In tissue-perfusion and vascular-transport settings, frameworks like DuMuX provide mature conservative FVM solvers on real vascular networks (Koch et al., 2020; 2021), but these finite volume-based methods face strict stability limits due to the underlying physics of the problem, and lack a GPU-based stochastic alternative tailored to metric graphs.

In this work, we bridge this gap by introducing the first algorithm (Algorithm 1) for simulating Brownian motions and Langevin diffusions on metric graphs. Our approach leverages a timestep splitting Euler-Maruyama discretization of the underlying SDE, which simultaneously resolves evolution along edges and transitions at vertices. We provide theoretical guarantees on this scheme's runtimes and consistency with the underlying SDE as the timestep approaches zero.

An important computational insight is the algorithm's parallelizability and well-suitedness to current modern GPU architectures. We implement it as a custom memory-aware CUDA kernel with Python bindings, enabling fast GPU-accelerated simulations that scale to large particle counts while effectively utilizing hardware capabilities. This implementation advances the practical utility of metric graph analyses and provides a first step toward computationally efficient stochastic simulations of these domains in high-performance computing environments.

We demonstrate that our method significantly outperforms a baseline finite volume scheme (DuMuX) on both a toy problem (star graphs) and a realistic problem – tracer transport in a cortical vascular network derived from a DuMuX tissue-perfusion model.

## 1.1 OUTLINE

In Section 1.2, we summarize our contributions. In Section 2, we provide the necessary background on metric graphs and Brownian motions on metric graphs. In Section 3, we present our main algorithm for simulating a Brownian motion on a metric graph with implementation details in Section 3.1. In Section 4, we present numerical results on star metric graphs and a real vascular network.

## 1.2 CONTRIBUTIONS

- We propose Algorithm 1, a timestep splitting Euler-Maruyama based discretization of the SDE of the Brownian motion, which is the first algorithm that we know of for simulating a Brownian motion and sampling on a metric graph. We provide an extension of this algorithm to general metric graphs in Algorithm 2.

- We show in Theorem 2 that the number of time-step splittings in Algorithm 1 is finite with high probability. Additionally, we show in Corollary 1 that the exit probabilities of the simulated particle using this algorithm converge to the vertex-edge jump probabilities of the underlying SDE as the timestep goes to zero.

- We provide fast, memory-aware implementations of Algorithm 1 and Algorithm 2 for GPUs in the form of a custom CUDA kernel with Python bindings and show significant speedups (up to ∼8000x faster on star metric graphs) over a GPU implementation using PyTorch (Paszke et al., 2019).

- We corroborate our theoretical results with numerical experiments on synthetic star metric graphs, significantly outperforming a baseline finite volume scheme in accuracy (Figure 3) and speed (Figure 4).

- We further validate our approach on general metric graphs by benchmarking on a real cortical vascular network derived from a DuMuX tissue-perfusion model for simulating tracer transport (Koch et al., 2020; 2021), matching a conservative finite-volume baseline while remaining stable at larger timesteps and achieving substantial speedups of up to ∼1500x on the drift-driven tracer task (Figure 5).

The code for our implementation is uploaded as part of the supplementary material.

## 2 BACKGROUND

### 2.1 METRIC GRAPHS

In this section, we provide some formal background on metric graphs.

**Definition 1** (Metric Graph). *Let $G = (V, E, l)$ be an n-node, m-edge, connected, oriented graph. We associate the line segment $(0, l_e)$ with each edge $e \in E$. We identify the endpoints of the interval $0$ and $l_e$ with the corresponding vertices of the edge, which we denote $e_{init}$ and $e_{term}$. The union of open metric edges associated with $G$ is defined as $\mathbf{\Gamma}^o := \{(e, x) \mid e \in E, x \in (0, l_e)\}$, and the union of closed metric edges as $\mathbf{\Gamma}^c := \{(e, x) \mid e \in E, x \in [0, l_e]\}$. The metric graph associated with $G$ is defined as $\mathbf{\Gamma} := V \cup \mathbf{\Gamma}^o$.*

Additionally, we allow edges to be semi-infinite, i.e., $l_e = \infty$. In this case, the terminal vertex of these edges is a vertex at infinity, and the intervals corresponding to these edges are $[0, \infty)$.

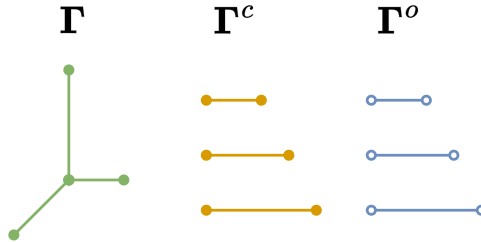

Figure 1: An example metric graph $\mathbf{\Gamma}$ and its associated spaces.

We also define a special case of metric graphs called *star metric graphs* where the graph has a single vertex and all edges are semi-infinite. The remainder of this paper will focus on star metric graphs, though all results extend to general metric graphs.

**Definition 2** (Star Metric Graph). *A star metric graph is a metric graph with a single vertex $v$, and all edges in $E$ are semi-infinite and have length $l_e = \infty$.*

We define the set of edges incident to a vertex $v \in V$ as

$$\mathcal{E}(v) := \{e \in E \mid e_{\text{init}} = v \text{ or } e_{\text{term}} = v\}.$$

### 2.1.1 FUNCTION SPACES ON METRIC GRAPHS

The metric structure of each edge combined with the discrete graph metric on $G$ leads to a natural definition of the distance $d : \mathbf{\Gamma} \times \mathbf{\Gamma} \to \mathbb{R}_+$ between two points on the metric graph. For $x, y \in \mathbf{\Gamma}$, let $\widetilde{G} = \left(\widetilde{V}, \widetilde{E}, \widetilde{l}\right)$ be the discrete graph obtained by adding two new vertices $x$ and $y$ to $G$ and splitting the edges on which they lie appropriately. Then we define the distance $d(x, y)$ as the

length of the shortest path between $x$ and $y$ in $\widetilde{G}$. This metric allows us to define the space $\mathcal{C}^k(\mathbf{\Gamma})$ as the space of functions on $\mathbf{\Gamma}$ that are $k$ times continuously differentiable.

In addition to the global metric structure of $\mathbf{\Gamma}$, the metric structure on each edge allows us to define a broader class of continuous functions by considering continuity restricted to the edges. For a function $f : \mathbf{\Gamma} \to \mathbb{R}$ (and also for functions $f : \mathbf{\Gamma}^c \to \mathbb{R}$), we define $f_e : [0, l_e] \to \mathbb{R}$ to be the restriction of $f$ to the closed edge $e$. We similarly define the restriction to open edges for functions $f : \mathbf{\Gamma}^o \to \mathbb{R}$.

We define the function space $\mathcal{C}^k(\mathbf{\Gamma}^o)$ as the space of functions on $\mathbf{\Gamma}$ whose restriction to each open edge $(0, l_e)$ is $k$ times continuously differentiable. Note that this can be naturally extended to functions on $\mathbf{\Gamma}^c$ by extending the restrictions to have values at the endpoints of the edges as: $f_e(0) := \lim_{x \to 0^+} f_e(x)$ and $f_e(l_e) := \lim_{x \to l_e^-} f_e(x)$. By a slight abuse of notation, we will also allow the use of $f_e(e_{\text{init}}) = f_e(0)$ and $f_e(e_{\text{term}}) = f_e(l_e)$ to denote these endpoint values.

A useful observation is to note that by identification of the vertices, for two edges $e, e' \in E$ that share a vertex $v \in V$ such that $e_{\text{init}} = e'_{\text{init}} = v$, we have for $f \in \mathcal{C}(\mathbf{\Gamma})$ that

$$f_e(0) = f_{e'}(0).$$

Similar results hold for different combinations of initial and terminal vertices of the edges. However, this is not the case for functions in $\mathcal{C}^k(\mathbf{\Gamma}^c)$ or $\mathcal{C}^k(\mathbf{\Gamma}^o)$. Specifically, for $f \in \mathcal{C}^k(\mathbf{\Gamma}^c)$, it need not be the case that $f_e^{(j)}(0) = f_{e'}^{(j)}(0)$ for edges $e, e' \in E$ that share an initial vertex $v \in V$, where $f_e^{(j)}$ denotes the $j$-th derivative of $f_e$.

Finally, for notational convenience, we define the notion of an *inward derivative* along an edge at a vertex that is independent of the orientation of the edge.

**Definition 3.** *Let $f \in \mathcal{C}^1(\mathbf{\Gamma})$. We define the inward derivative of $f$ at a vertex $v \in V$ along an edge $e \in E$ incident to $v$ as*

$$\partial_e f(v) := \begin{cases} -\frac{\partial f_e}{\partial x}(0) & \text{if } e_{init} = v, \\ \frac{\partial f_e}{\partial x}(l_e) & \text{if } e_{term} = v. \end{cases}$$

*Note that flipping the orientation of edge $e$ does not change the sign of the inward derivative. See Figure 2 for a visual depiction of the inward derivative.*

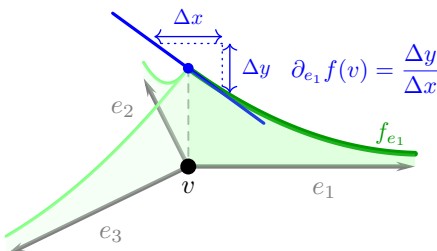

Figure 2: Visual depiction of the inward derivative $\partial_{e_1} f(v)$ along an edge $e_1$ at a vertex $v$. Its sign is independent of the orientation of the edge.

## 2.2 Brownian motions on Metric Graphs

Brownian motions on metric graphs are extensively studied in (Kostrykin et al., 2012). A Brownian motion on a metric graph is generated by the standard second-order generator of the Brownian motion restricted to each open edge, along with specific boundary conditions at the vertices known as gluing conditions.

Let $\mu \in \mathcal{C}^1(\mathbf{\Gamma}^c)$ and $\sigma \in \mathcal{C}^1(\mathbf{\Gamma})$ be functions that denote drift and diffusion coefficients respectively. The generator $\mathcal{L}$ of a Brownian motion on the metric graph applied to a function $f \in \mathcal{C}^2(\mathbf{\Gamma})$ is given by

$$(\mathcal{L}f)_e = \mathcal{L}_e f_e \quad \text{for all } e \in E \tag{1}$$

where $\mathcal{L}_e$ is the generator of a Brownian motion on the open edge $e$:

$$\mathcal{L}_e f_e(x) := \frac{\partial f_e(x)}{\partial x} \mu_e(x) + \frac{1}{2} \sigma_e^2(x) \frac{\partial^2 f_e(x)}{\partial x^2}. \tag{2}$$

The domain of $\mathcal{L}$ is restricted to functions $f \in \mathcal{C}^2(\mathbf{\Gamma})$ that satisfy a set of gluing boundary conditions at each vertex $v \in V$. (Kostrykin et al., 2012) shows that a class of gluing conditions called the *Wentzell boundary conditions* characterizes all possible Brownian motions. The Wentzell boundary conditions for $f \in \mathcal{C}^2(\mathbf{\Gamma})$ are given by

$$a_v f(v) - \sum_{e \in \mathcal{E}(v)} b_{ve} \partial_e f(v) + \frac{1}{2} c_v f''(v) = 0 \quad \text{for all } v \in V \tag{3}$$

where $a_v \in [0,1), b_{ve} \in [0,1], c_v \in [0,1]$ are constants that satisfy

$$a_v + \sum_{e \in \mathcal{E}(v)} b_{ve} + c_v = 1 \quad \text{for all } v \in V. \tag{4}$$

In this paper, we will consider the case where $a_v = 0, c_v = 0$, which are often referred to as the *standard* boundary conditions.

**Definition 4** (Standard Boundary Conditions). *$f \in \mathcal{C}^2(\mathbf{\Gamma})$ satisfies the standard boundary conditions if*

$$\sum_{e \in \mathcal{E}(v)} b_{ve} \partial_e f(v) = 0 \quad \text{for all } v \in V. \tag{5}$$

*where $b_{ve} \in [0,1]$ are constants that satisfy*

$$\sum_{e \in \mathcal{E}(v)} b_{ve} = 1 \quad \text{for all } v \in V. \tag{6}$$

For convenience, we define the simplex

$$\Delta_v := \left\{ x \in \mathbb{R}^{\mathcal{E}(v)} \mid x_e \in [0,1] \text{ and } \mathbf{1}^T x = 1 \right\}$$

and note that $b_v \in \Delta_v$ defines a vertex-edge jump probability distribution at each vertex $v \in V$.

The Brownian motion generated by the generator with standard boundary conditions is conservative, and an extensive analysis of the stochastic properties is provided in (Freidlin & Sheu, 2000). In particular, (Freidlin & Sheu, 2000) derives an SDE for this Brownian motion and characterizes the behavior of the process at the vertices of the metric graph. As a first simplification, we only need to characterize the stochastic process's behavior at a single vertex since this is a local property that can be extended to other vertices. Effectively, we only need to consider the behavior of the process on star metric graphs. We restate the main results of (Freidlin & Sheu, 2000) in the following theorem.

**Theorem 1** (Lemma 2.2 and Corollary 2.4 in (Freidlin & Sheu, 2000)). *Let $X_t = (e_t, x_t)$ be a Brownian motion on a star metric graph $\mathbf{\Gamma}$ with standard boundary conditions. There exists a 1-dimensional Brownian motion $W_t$ and a local time process $l_t$ adapted to the filtration generated by $X_t$ such that*

$$dx_t = \mu_{e_t}(x_t) \, dt + \sigma_{e_t}(x_t) \, dW_t + dl_t. \tag{7}$$

*Moreover, the local time process $l_t$ is a continuous, non-decreasing process that only increases when the particle is at the vertex, i.e., $x_t = 0$.*

*Let $\tau_\delta := \inf \{t \geq 0 : x_t = \delta\}$ be the first time the process exits a ball of radius $\delta$ centered at the vertex (assume $X_0 = v$). The discrete edge process $e_t$ is characterized by the following transition probabilities,*

$$\lim_{\delta \to 0} \mathbb{P}[e_{\tau_\delta} = i] = b_{vi} \quad \text{for all } i \in \mathcal{E}(v). \tag{8}$$

## 3 Timestep Splitting Euler-Maruyama Scheme for Metric Graphs

In this section, we present our main algorithm, an Euler-Maruyama-based method for simulating Brownian motion on a metric graph via timestep splitting. First, we recall the standard Euler-Maruyama discretization for a particle on the real line $\mathbb{R}$ with the update rule

$$X_{k+1} = X_k + \mu\left(X_k\right)\Delta t + \sigma\left(X_k\right)W_{k+1}\sqrt{\Delta t}, \tag{9}$$

where $W_k$ are i.i.d. standard normal random variables. Note that (9) is a first-order finite difference approximation with timestep $\Delta t$ of the SDE

$$dX_t = \mu\left(X_t\right)dt + \sigma\left(X_t\right)dW_t, \tag{10}$$

where $W_t$ is a standard Brownian motion.

We extend the Euler-Maruyama method to simulate Brownian motions on a metric graph $\boldsymbol{\Gamma}$. The main challenge in implementing a discretization scheme for the SDE (7) is to handle the case when the particle crosses a vertex in one Euler-Maruyama step in a way that is consistent with the underlying Brownian motion. To tackle this scenario, we propose a timestep splitting approach that first performs a partial Euler-Maruyama step so that the particle is exactly at the vertex and then chooses a new edge based on the vertex-edge jump probabilities $b_v$. Following this, we complete the remaining Euler-Maruyama step using the drift and diffusion coefficients of the new edge.

A complication that arises is that the remaining step could also result in a vertex crossing. A recursive application of the timestep splitting approach allows us to handle multiple vertex crossings in a single time step. The detailed algorithm in the case of a single vertex is described in Algorithm 1. A visual depiction of the algorithm is shown in Figure 6. An explicit general-graph variant that handles finite edge lengths and both endpoints is given in Section A.5. We focus on star metric graphs in the main text for clarity.

A possible issue with the algorithm is that the number of timestep splittings required to simulate a single step of the Brownian motion is not guaranteed to be finite. This could lead to an infinite runtime for simulating a finite timestep. However, we rigorously establish in Theorem 2 that this scenario does not arise, ensuring that the algorithm terminates in a finite number of steps with high probability.

**Theorem 2** (Finite vertex crossings with high probability). *Let $M$ be the number of vertex crossings the particle makes in a single Euler-Maruyama step starting from the vertex, as computed in Algorithm 1 with input $(e, 0)$. Then for all $k > 0$, we have*

$$\mathbb{P}\left[M \le k\right] \ge 1 - e^{-\frac{(k-\gamma)^2}{4k}}$$

*where $\gamma$ is defined as the following dimensionless quantity:*

$$\gamma := \Delta t \cdot \max_{e \in \mathcal{E}(v)} \frac{\mu_e^2(v)}{\sigma_e^2(v)}.$$

In addition to the above, we also show that the exit probabilities of the simulated particle using this partial stepping algorithm converge to the vertex-edge jump probabilities of the SDE as the timestep goes to zero. Intuitively, this is because the number of vertex crossings approaches 1 as the timestep goes to zero. We formalize this in Theorem 3 and Corollary 1.

**Theorem 3** (Number of crossings is 1 with high probability). *Let $M$ be the number of vertex crossings the particle makes in a single Euler-Maruyama step starting from the vertex, as computed in Algorithm 1 with input $(e, 0)$. Let $\gamma$ be defined as in Theorem 2. Then,*

$$\mathbb{P}\left[M = 1\right] \ge \Omega\left(e^{-\gamma}\right).$$

*As a consequence, we can choose*

$$\Delta t \le \mathcal{O}\left(\frac{1}{\max_{e \in \mathcal{E}(v)} \frac{\mu_e^2(v)}{\sigma_e^2(v)}} \log\left(\frac{1}{1-\delta}\right)\right)$$

*to ensure that $\mathbb{P}\left[M = 1\right] \ge 1 - \delta$.*

---

**Algorithm 1** Timestep Splitting Euler-Maruyama Algorithm for Metric Graphs

---

**Require: Star** metric graph $\mathbf{\Gamma} = (V, E, l)$ (star graph so $V = \{v\}$ is a singleton and all edges are semi-infinite, with $e_{\text{init}} = v \quad \forall e \in E$), drift function $\mu : \mathbf{\Gamma}^c \to \mathbb{R}$, diffusion function $\sigma : \mathbf{\Gamma} \to \mathbb{R}_+$, edge-vertex jump probabilities $b_v \in \Delta_E$.
1: **procedure** EM-STEP$(e, X, \Delta t)$          $\triangleright$ $(e, X) \in \mathbf{\Gamma}$, $\Delta t$ is the time to simulate
2:     $M \leftarrow 0$.          $\triangleright$ Number of vertex crossings
3:     **if** $X \neq 0$ **then**          $\triangleright$ Particle is not at the vertex
4:        Sample $W \sim \mathcal{N}(0, 1)$.
5:        $\widetilde{X} \leftarrow X + \mu_e(X)\Delta t + \sigma_e(X)\sqrt{\Delta t}W$.
6:        **if** $\widetilde{X} < 0$ **then**          $\triangleright$ Particle hits vertex
7:           Solve $X + \alpha\mu_e(X)\Delta t + \sigma_e(X)\sqrt{\alpha\Delta t}W = 0$ for $\alpha$.
8:           Sample $\widetilde{e}$ from $\mathcal{E}(v)$ according to $b_v$.
9:           $e \leftarrow \widetilde{e}$.
10:         $X \leftarrow 0$.
11:         $\Delta t \leftarrow (1 - \alpha)\Delta t$.
12:        **else**
13:           **return** $\left(e, \widetilde{X}\right)$.
14:     **while** $\Delta t > 0$ **do**          $\triangleright$ Particle is at vertex
15:        $M \leftarrow M + 1$.
16:        Sample $W \sim \mathcal{N}(0, 1)$.
17:        $\widetilde{X} \leftarrow \widetilde{X}_1 + \mu_e(0)\Delta t + \sigma_e(0)\sqrt{\Delta t}|W|$.
18:        **if** $\widetilde{X} < 0$ **then**          $\triangleright$ Particle hits vertex again
19:           Sample $\widetilde{e}$ from $\mathcal{E}(v)$ according to $b_v$.
20:           $e \leftarrow \widetilde{e}$.
21:           $\alpha \leftarrow \frac{W^2\sigma_e^2(0)}{\mu_e^2(0)\Delta t}$.
22:           $\Delta t \leftarrow (1 - \alpha)\Delta t$.
23:        **else**
24:           **return** $\left(e, \widetilde{X}\right)$.

---

**Corollary 1** (Jump probabilities converge to $b_v$)**.** *Let $(\widetilde{e}, X)$ be the output of the procedure in Algorithm 1 with timestep $\Delta t$ and input $(e, 0)$. Then,*

$$\lim_{\Delta t \to 0} \mathbb{P}[\widetilde{e} = i] = b_{vi} \quad \text{for all } i \in \mathcal{E}(v).$$

Proofs of Theorem 2, Theorem 3, and Corollary 1 can be found in Section A.2.

### 3.1 FAST, PARALLELIZED, MEMORY-AWARE IMPLEMENTATION IN CUDA

The standard Euler-Maruyama discretization lends itself to a fast, parallelized implementation on GPUs because each particle can be simulated independently. Previous works have explored the use of GPUs for algorithms like Markov Chain Monte Carlo and Gibbs sampling in Euclidean spaces (Sountsov et al., 2024; Quiroz et al., 2015; Terenin et al., 2019). Our Algorithm 1 enjoys similar computational benefits, and we can leverage the parallelism of GPUs to simulate a large number of particles in parallel on metric graphs. Further, this algorithm works particularly well with GPUs' exact architecture, where the balance between memory transfers and compute operations significantly impacts practical performance.

We provide a brief overview of the architectural details of GPUs that are relevant to our implementation; further details can be found in the CUDA programming guide (Nvidia, 2011). A GPU consists of thousands of CUDA cores that can independently execute threads of computation in parallel. Along with a large number of compute units, the GPU also has a hierarchy of memory that the cores can access. The hierarchy stems from a fundamental trade-off between memory size, latency, and bandwidth. The fastest memory is the register memory, which is local to each thread and is used to store intermediate results. The next level of memory is the shared memory,

which is shared between a local group of threads. The global memory is the largest and slowest memory, but it is accessible by all threads.

Achieving high performance on GPUs requires optimizing memory accesses so that the threads can maximize the utilization of the compute units. A significant advantage of Monte Carlo methods, like the standard Unadjusted Langevin Algorithm (ULA) as well as our Algorithm 1, is that they lend themselves to highly optimized memory access patterns. Specifically, since each particle simulation is completely independent, we can assign each thread to simulate a single particle. This allows the particle's state to remain in register memory over multiple timesteps, which is the fastest memory available. Consequently, the compute units can operate at peak utilization without being bottlenecked by memory accesses. Slower transfers to and from global memory are only required when evaluating ensemble statistics, such as histogram averages. We implement our algorithm in CUDA to take advantage of these architectural features. Specifically, we provide CUDA kernels for running multiple timesteps of Algorithm 1 for multiple particles in parallel. We also provide a CUDA kernel for computing empirical histograms of these particles, which allows us to measure the error between their density and the steady-state density. We present detailed numerical experiments in Section 4. CUDA kernel source code can be found in Section A.6.

### 3.2 Extension to general metric graphs

Algorithm 1 extends directly from star graphs to general metric graphs by storing, for each vertex, the incident edge indices, their orientations, and cumulative jump weights. The timestep-splitting procedure is unchanged on general graphs: each edge has finite length, bounces can occur at either endpoint, and the outgoing edge is sampled from the jump distribution of the hit vertex. We explicitly check both ends, split the step to the hit point, and recurse on the remaining time. Further details can be found in Section A.5, and the pseudocode is given in Algorithm 2.

## 4 Numerical Experiments

### 4.1 Star metric graphs

We consider a simple star metric graph with 5 edges and 1 vertex. For simplicity, we choose constant diffusion $\sigma_e(x) = \sigma$ for all edges $e \in E$. We choose the vertex-edge jump probabilities to be uniform, i.e., $b_{vi} = \frac{1}{5}$. We consider two cases of drift, driven by a linear potential and by a quadratic potential.

**Linear Potential** In the case of linear potentials, each edge has constant drift towards the vertex with varying magnitudes given by $\mu_{e_i}(x) = -10 \cdot i$ for $i \in \{1, 2, 3, 4, 5\}$. This drift corresponds to a linear potential with constant diffusion along each edge, which is equivalent to the Ornstein-Uhlenbeck (Ornstein, 1930) process on each edge, but edges interact through the gluing boundary conditions. The steady-state distributions on each edge are exponential: $\rho_i(x) = B \exp(-\frac{\mu_i}{D}x)$ for $x \in [0, \infty)$, with $B = \frac{D}{\sum_i \frac{1}{\mu_i}}$ so the total mass is 1. Note that all edges have the same normalizing constant $B$ due to continuity of density at the vertex.

**Quadratic Potential** In the case of quadratic potentials, the drift is given by $\mu_{e_i}(x) = -10 \cdot i \cdot x$ for $i \in \{1, 2, 3, 4, 5\}$. The steady-state distributions are Gaussian: $\rho_i(x) = B \exp(-\frac{\mu_i}{2D}x^2)$ for $x \in [0, \infty)$, with $B = \frac{\sqrt{2/(D\pi)}}{\sum_i 1/\sqrt{\mu_i}}$.

We run Algorithm 1 in parallel for multiple particles over multiple timesteps. We measure the error in the particles' density (after sufficient simulation time) with respect to the analytical steady-state density. As a baseline, we compare this with the density obtained by solving the Fokker-Planck equation using a Finite Volume Method (FVM) scheme. Numerical results are presented in Figure 3, and runtime comparisons between different implementations are in Figure 4. Further details are provided in the appendix.

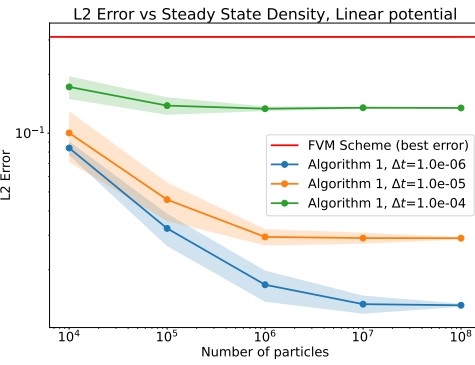
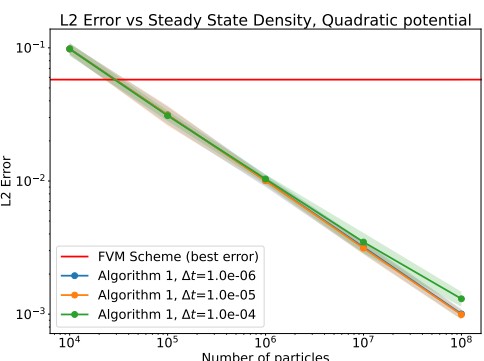

Figure 3: Error in density estimation for linear and quadratic potentials. The FVM scheme directly solves the Fokker-Planck equation to obtain the steady-state density. We compare the *best case* error (over discretization parameters) of this scheme with the error obtained by running Algorithm 1 for multiple particle counts and values of the timestep. We estimate the density using a simple histogram with a bin size equal to the discretization of the FVM scheme. The error is computed as the empirical L2 distance between the estimated density and the analytical steady-state density. We observe that Algorithm 1 results in significantly lower error compared to the FVM scheme for the same level of spatial and temporal discretizations.

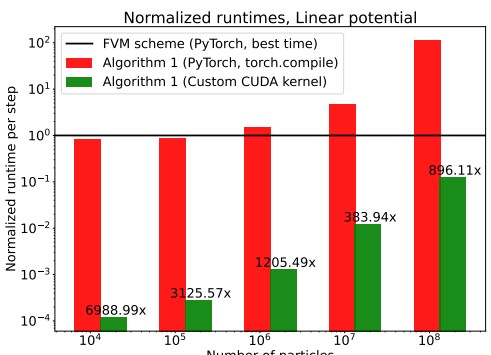
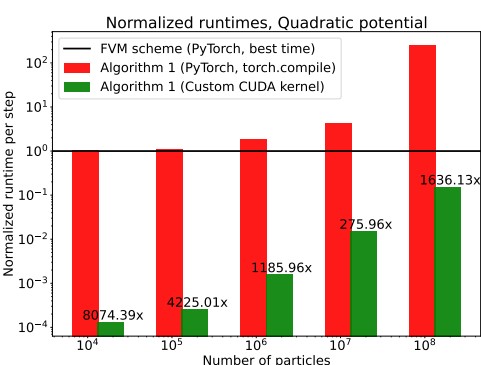

Figure 4: Normalized runtimes per step, aggregated over different discretization parameters for Algorithm 1 for linear and quadratic potentials compared with the best runtime for the FVM scheme. We observe that the FVM scheme has a significantly higher runtime compared to Algorithm 1 for the same level of spatial and temporal discretizations. Additionally, our custom CUDA kernel for Algorithm 1 is significantly faster (up to ∼8000x speedup) than the PyTorch implementation (speedups indicated on the bars). We observe slightly higher runtimes for the linear potential, which is expected due to the increased likelihood of vertex crossings per timestep. All experiments were run on an NVIDIA RTX A6000 GPU.

## 4.2 Tracer transport on a real vascular network

We evaluate Algorithm 1 on a mouse cortical microvascular network from the DuMuX `embedded_network_1d3d` example (Koch et al., 2020; 2021; Blinder et al., 2013), represented as a one-dimensional FoamGrid with 1,458 vertices, 1,470 edges, max degree 4, and rich cyclic structure. Using boundary conditions estimated by (Schmid et al., 2017) and the pressure and network data available from (Schmid, 2017), we can simulate blood flow using DuMuX's built-in solvers. DuMuX also provides a conservative finite-volume discretization for tracer transport on this graph; we treat it as a deterministic baseline and mirror its layout and boundary conditions.

**Flow-driven drift** We first solve for blood flow in the DuMuX `embedded_network_1d3d` tissue-perfusion setup on a $200\,\mu\mathrm{m}^3$ cortical subvolume to obtain per-edge volume fluxes $Q_e$ and cross-

sectional areas $A_e$ by solving a Darcy flow problem in the tissue coupled to 1D flow in the vessels. We set the drift velocity on each edge to the centerline value $u_e = Q_e/A_e$ and simulate advection–diffusion of a passive tracer with both the DuMuX FVM and Algorithm 2 on this nontrivial network. This setting is widely studied for example in understanding the transport of oxygen and nutrients in cerebral microvasculature (Gagnon et al., 2016). An analytic solution is unavailable, so we compare tracer fields qualitatively and emphasize runtime and stability. Figure 5 shows runtime comparisons and a steady state tracer density visualization on the vascular network. Our algorithm matches the large-scale tracer patterns, runs significantly faster than the DuMuX FVM baseline, and remains stable for timestep sizes beyond the FVM stability limit.

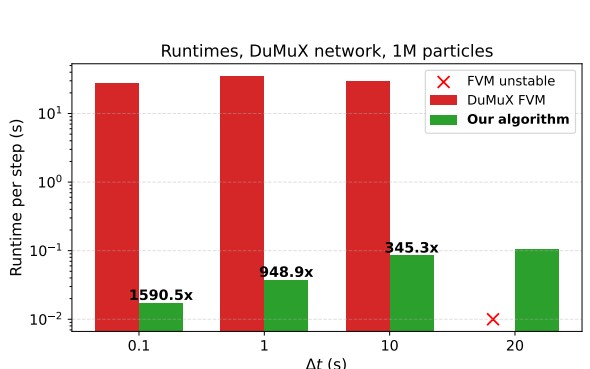
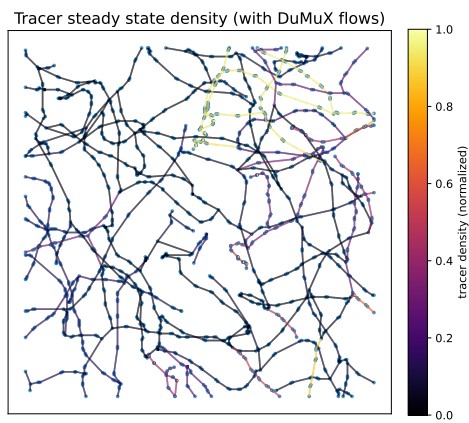

Figure 5: Simulation of tracer transport as advection–diffusion on a real cortical vascular network, with runtime scaling (left) and steady state tracer density visualization on the network (right). We observe that Algorithm 2 achieves up to ~1500x speedup over the DuMuX FVM scheme, while remaining stable beyond the FVM stability limit for large timesteps where FVM fails to converge. This demonstrates the practical utility of our algorithm on real-world metric graphs. These experiments were run on an NVIDIA RTX 4090 GPU.

## 5 Conclusion, Limitations and Future Work

In this paper, we presented a novel Euler-Maruyama-based algorithm for simulating Brownian motions on metric graphs. Our algorithm uses a timestep splitting approach that allows us to handle vertex crossings in a way that is consistent with the underlying Brownian motion. We rigorously established that the number of vertex crossings is finite with high probability and that the exit probabilities of the simulated particle converge to the vertex-edge jump probabilities of the SDE as the timestep goes to zero. We also provided a fast, parallelized, memory-aware implementation in CUDA that takes advantage of the architecture of modern GPUs. We demonstrated the effectiveness of our algorithm through numerical experiments on simple star metric graphs as well as a real vascular network.

Promising future directions include developing higher-order variants of timestep splitting algorithms like Algorithm 1 for simulating Brownian motions on metric graphs. Further, bringing existing sampling algorithms inspired by optimization perspectives (Chewi, 2023) like proximal sampling (Liang & Chen, 2022), mirror Langevin (Hsieh et al., 2018), and the Metropolis-adjusted Langevin algorithm (Xifara et al., 2014) to the domain of metric graphs would serve as an interesting research direction. On the theoretical side, developing non-asymptotic convergence rates for these algorithms is a potential avenue for future work. Finally, extending the optimized CUDA implementation to accommodate interacting systems by exploiting shared memory and other architectural features of GPUs is another promising direction.

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

# A APPENDIX

## A.1 VISUAL REPRESENTATION OF ALGORITHM 1

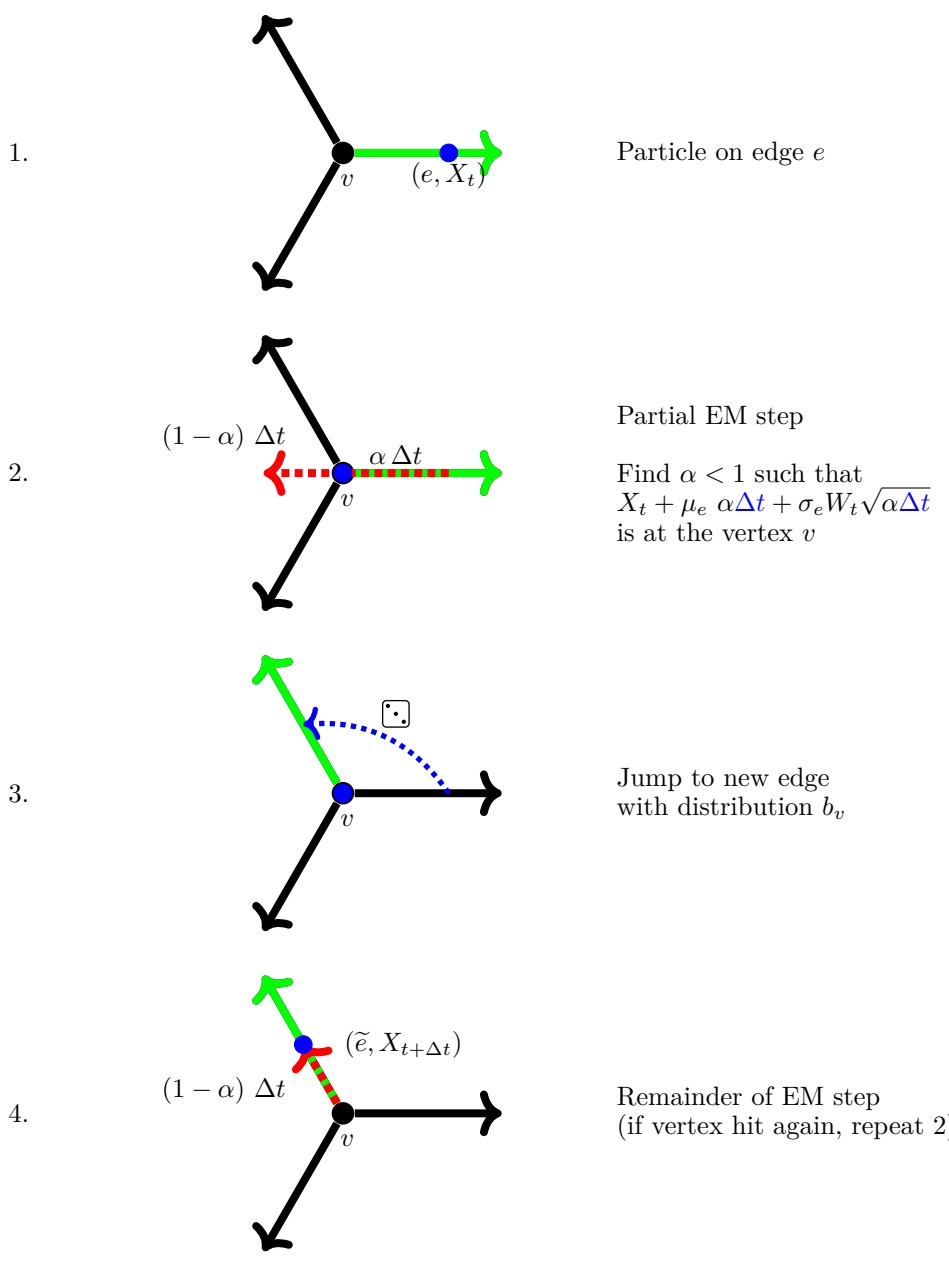

Figure 6: Four stages of one iteration of the timestep splitting Euler-Maruyama algorithm. The particle starts by taking a standard Euler-Maruyama step on the edge it is currently on. If it crosses the vertex, the standard step is split into a partial step until it hits the vertex. Then, it samples a new edge from the vertex-edge jump probabilities and continues the remainder of the partial step on the new edge. If it crosses the vertex again, it repeats the process. This timestep splitting process continues until the particle no longer crosses the vertex in a partial step.

## A.2 Proofs of Theorems

### A.2.1 Finite Vertex Crossings with High Probability

*Proof of Theorem 2.* Let $I_k$ for $k > 0$ be iid variables that take values in $\mathcal{E}(v)$ according to the distribution $b_v$. Let $W_k \sim \mathcal{N}(0, 1)$ for $k > 0$. We define the following sequence of random variables:

$$h_k := h_{k-1} - \frac{\sigma^2_{I_k}(v)}{\mu^2_{I_k}(v)} W_k^2,$$

where $h_0 := \Delta t$.

Define the stopping time $\tau := \inf\{k > 0 : h_k \leq 0\}$. First, observe that $M = \tau$. To see this, note that after one iteration of the loop in Algorithm 1, we have $\Delta t' = \Delta t - \frac{\sigma^2_{I_1}(v)}{\mu^2_{I_1}(v)} W_1^2$. If $\Delta t' \leq 0$, then $M = 1$, and if $\Delta t' > 0$, then we continue the loop with $\Delta t'$ in place of $\Delta t$. By induction, we see that $M = \tau$. We can define the following upper-bounding sequence by considering the worst case over all possible choices of $I_k$,

$$\widetilde{h}_k := \widetilde{h}_{k-1} - \frac{h_0}{\gamma} W_k^2, \text{ where } \widetilde{h}_0 = h_0 = \Delta t.$$

We can solve the recursion for $\widetilde{h}_k$ to get

$$\widetilde{h}_k = h_0 \left(1 - \frac{1}{\gamma} \sum_{i=1}^{k} W_i^2\right).$$

By construction, $\widetilde{h}_k \geq h_k$ for all $k \geq 0$. We define a similar stopping time for $\widetilde{h}$, as $\widetilde{\tau} := \inf\{k > 0 : \widetilde{h}_k \leq 0\}$. Clearly, $h_k \leq \widetilde{h}_k \implies \widetilde{\tau} \geq \tau$. Now, observe that $\sum_{i=1}^{k} W_i^2 = \chi_k^2$ is a chi-squared random variable with $k$ degrees of freedom. We have,

$$h_0 \left(1 - \frac{1}{\gamma} \chi_k^2\right) \leq 0 \iff \chi_k^2 \geq \gamma \implies \widetilde{\tau} \leq k \implies \tau \leq k.$$

Therefore,

$$\mathbb{P}[M \leq k] = \mathbb{P}[\tau \leq k] \geq \mathbb{P}[\widetilde{\tau} \leq k] \geq \mathbb{P}[\chi_k^2 \geq \gamma].$$

To control the tail of $\chi_k^2$, we use the following bound from Lemma 1 in Section 4.1 of (Laurent & Massart, 2000),

$$\mathbb{P}\left[k - \chi_k^2 \geq 2\sqrt{kx}\right] \leq e^{-x}.$$

We have,

$$\mathbb{P}\left[\chi_k^2 \leq k - 2\sqrt{kx}\right] \leq e^{-x}$$
$$\implies 1 - \mathbb{P}\left[\chi_k^2 \geq k - 2\sqrt{kx}\right] \leq e^{-x}.$$

By setting $x = \frac{(k-\gamma)^2}{4k}$, we get

$$\mathbb{P}\left[\chi_k^2 \geq \gamma\right] \geq 1 - e^{-\frac{(k-\gamma)^2}{4k}}.$$

This completes the proof. □

### A.2.2 Number of Crossings is 1 with High Probability

*Proof of Theorem 3.* We use the same notation defined in the proof of Theorem 2.

First, observe that $M = 1 \implies W_1^2 \geq \gamma$.

So, we have $\mathbb{P}[M = 1] \leq \mathbb{P}[W_1^2 \geq \gamma]$.

Since $W_1$ is a standard normal random variable, we have

$$\mathbb{P}\left[W_1^2 \geq \gamma\right] = \mathbb{P}\left[|W_1| \geq \sqrt{\gamma}\right] = 2\left(1 - \Phi\left(\sqrt{\gamma}\right)\right),$$

where $\Phi$ is the CDF of the standard normal distribution.

We use a standard lower bound on the CDF (Casella, 2001) of the normal distribution to get

$$1 - \Phi\left(\sqrt{\gamma}\right) \geq C \exp\left(-\frac{\gamma}{2}\right)$$

$$\implies \mathbb{P}\left[M = 1\right] \geq C \exp\left(-\frac{\gamma}{2}\right) \geq \Omega\left(e^{-\gamma}\right).$$

This completes the proof.

$\square$

### A.2.3 Jump Probabilities Converge to $b_v$

*Proof of Corollary 1.* Clearly, if $M = 1$, then $\widetilde{e} = i$ with probability $b_{vi}$, since $\widetilde{e}$ is sampled from the distribution $b_v$ once. By noting that $\gamma \to 0$ as $\Delta t \to 0$, we can apply Theorem 3 to conclude that $\mathbb{P}\left[M = 1\right] \to 1$ as $\Delta t \to 0$. This completes the proof. $\square$

### A.3 Baseline Finite Volume Scheme

We provide a brief overview of the Finite Volume Method (FVM) scheme for solving the Fokker-Planck equation on metric graphs. On each edge $e \in E$, we discretize the Fokker-Planck equation using a standard FVM scheme; see (LeVeque, 2002) for details. We use an upwinding scheme to discretize the drift term and a central difference scheme to discretize the diffusion term. We use a first-order explicit Euler scheme to discretize the time derivative.

We now provide details of the flux balance condition at the vertex. Denote the density of the cell adjacent to the vertex on edge $i$ by $\rho_i$. To mimic the gluing boundary conditions, we use a flux distribution at the vertex that is proportional to the jump probabilities $b_v$. Specifically, let $F_{ij}$ be the flux from edge $i$ to edge $j$ at the vertex. We decompose the flux into a drift and diffusion component.

**Drift Component** Since we use an upwinding scheme to discretize the drift term, the drift component of the flux from edge $i$ to edge $j$ is zero if the drift $\mu_i(v)$ is away from the vertex on edge $i$. If the drift is towards the vertex, then the drift component of the flux is given by

$$F_{ij}^{\text{drift}} = \mu_i(v)\rho_i/b_{vi} \frac{b_{vj}}{\sum_{k \neq i} b_{vk}}.$$

Note that the drift flux is distributed to all target edges, proportional to the jump probabilities. The density from the source is normalized by the jump probability to account for the fact that the density is distributed to all target edges.

Intuitively, the jump probability can be interpreted as the relative "cross-sectional areas" of the edges at the vertex. The density is the linear density of the particles on the edge. When computing fluxes across different edges, we need to account for the relative "cross-sectional areas" of the edges at the vertex. Hence, we normalize by the appropriate jump probabilities.

**Diffusion Component** The diffusion component of the flux is given by

$$F_{ij}^{\text{diffusion}} = \frac{\sigma(v)}{2}\left(\frac{\rho_i/b_{vi} - \rho_j/b_{vj}}{\Delta x}\right)b_{vj}$$

A similar normalization is applied to the density terms to account for the relative "cross-sectional areas" of the edges at the vertex.

The total flux into the cells at the vertex on each edge is the sum of all the incoming drift and diffusion components from every other edge.

### A.4 Steady-state normalizers for star graphs

For completeness, we collect the normalization constants used in the star-graph steady states in Section 4.

Linear drift ($\mu_{e_i}(x) = -10 \cdot i$): the steady-state density on edge $i$ is $\rho_i(x) = B \exp(-\frac{\mu_i}{D}x)$ for $x \in [0, \infty)$ with $B = \frac{D}{\sum_i \frac{1}{\mu_i}}$ to ensure total mass 1. Continuity at the vertex enforces the same $B$ across edges.

Quadratic drift ($\mu_{e_i}(x) = -10 \cdot i \cdot x$): the steady-state density is $\rho_i(x) = B \exp(-\frac{\mu_i}{2D}x^2)$ with $B = \frac{\sqrt{2/(D\pi)}}{\sum_i 1/\sqrt{\mu_i}}$. Continuity yields a common $B$.

### A.5 Extension of Algorithm 1 to general metric graphs

Algorithm 1 was presented for a star graph for clarity. The timestep-splitting procedure is unchanged on general graphs: each edge has finite length, bounces can occur at either endpoint, and the outgoing edge is sampled from the jump distribution of the hit vertex. We explicitly check both ends, split the step to the hit point, and recurse on the remaining time. The full pseudocode is given below in Algorithm 2.

---

**Algorithm 2** Timestep Splitting Euler–Maruyama on a General Metric Graph

---

**Require:** Metric graph $\mathbf{\Gamma} = (V, E, l)$, edge endpoints $(e_{\text{init}}, e_{\text{term}})$, drift $\mu$, diffusion $\sigma$, per-vertex jump cumulative weights $\{\text{cumw}[v]\}$ with edge indices $\{\text{adj}[v]\}$ and orientations $\{\text{orient}[v]\}$.

1: **procedure** EM-Step-General$(e, x, \Delta t)$            ▷ $x \in [0, l_e]$
2:      **while** $\Delta t > 0$ **do**
3:          Sample $W \sim \mathcal{N}(0, 1)$.
4:          $\widetilde{X} \leftarrow x + \mu_e(x)\Delta t + \sigma_e(x)\sqrt{\Delta t}\,W$.
5:          **if** $0 < \widetilde{X} < l_e$ **then return** $(e, \widetilde{X})$
                                                      ▷ no vertex hit
6:          hit_start $\leftarrow (\widetilde{X} \leq 0)$; $v \leftarrow$ hit_start?$e_{\text{init}} : e_{\text{term}}$.
7:          $d \leftarrow$ hit_start?$x : (l_e - x)$            ▷ distance to the hit vertex
8:          Solve $d + \alpha\,\mu_e(x)\Delta t + \sigma_e(x)\sqrt{\alpha\Delta t}\,W = 0$ for $\alpha \in [0, 1]$.
9:          $\Delta t \leftarrow (1 - \alpha)\Delta t$               ▷ remaining time
10:        Sample $j$ from the cumulative weights for $v$; $e \leftarrow \text{adj}[v][j]$; $o \leftarrow \text{orient}[v][j]$.
11:        $l_e \leftarrow \text{length}(e)$; $x \leftarrow (o = \text{'init'})?0 : l_e$     ▷ start at chosen end of new edge

---

### A.6 CUDA Kernels

We present source code for our CUDA kernels for running Algorithm 1 and Algorithm 2 for multiple particles over multiple timesteps. We also provide a kernel for computing empirical histograms of the particles. Python bindings and other code can be found in the uploaded supplementary material.

```cuda
// langevin-gpu/src/langevin_kernel.cu
#include "langevin_kernel.h"
#include <curand_kernel.h>

#define tol 1e-10f
#define max_iterations_per_step 100
#define steps_per_kernel 1000
#define potential_linear 0
#define potential_quadratic 1

extern "c" {
__device__ float compute_drift(const int potential_type, const int edge_index,
                               const float position,
                               const float *drift_coeffs) {
  const float coeff = drift_coeffs[edge_index];
  if (potential_type == potential_quadratic) {
    return coeff * position;
  }
  return coeff;
}

__device__ float solve_quadratic(float a, float b, float c) {
  // numerically stable solution to quadratic equation
  if (a == 0.0f) {
    return -c / b;
  }
  float discriminant = sqrtf(fmaxf(b * b - 4.0f * a * c, 0.0f));
  if (b > 0.0f) {
    return (-b - discriminant) / (2.0f * a);
  } else {
    return (2.0f * c) / (-b + discriminant);
  }
}

__global__ void
langevin_multi_step_kernel(int *edges, float *positions, int *bounces,
                           int *bounce_instances, const float *edge_lengths,
                           const float *jump_weights, const float *drift_coeffs,
                           const int potential_type, const float base_dt,
                           const float sigma, const int num_edges,
                           const int num_particles, curandstate *states) {
  const int tid = blockidx.x * blockdim.x + threadidx.x;
  if (tid >= num_particles)
    return;

  // printf("num_particles %d", num_particles);
  int edge = edges[tid];
  float x = positions[tid];
  int bounce_count = bounces[tid];
  int bounce_instance = bounce_instances[tid];
  curandstate local_state = states[tid];
```

```
972      if (edge < 0 || edge >= num_edges) {
973        printf("invalid initial edge %d for particle %d\n", edge, tid);
974        edge = 0;
975      }
976
977      for (int step = 0; step < steps_per_kernel; ++step) {
978        float dt = base_dt;
979        int iterations = 0;
980
981        while (dt > 0.0f && iterations++ < max_iterations_per_step) {
982          float w = curand_normal(&local_state);
983
984          float drift = compute_drift(potential_type, edge, x, drift_coeffs);
985          float sqrt_dt = sqrtf(dt);
986          float x_next = x + dt * drift + sigma * sqrt_dt * w;
987          float current_length = edge_lengths[edge];
988
989          if (current_length <= 0.0f) {
990            printf("invalid edge length %f for edge %d\n", current_length, edge);
991            current_length = 1.0f;
992          }
993
994          if (x_next > 0.0f && x_next <= current_length) {
995            // no bounce
996            x = x_next;
997            dt = 0.0f;
998          } else if (x_next > current_length) {
999            // also no bounce
1000           x = 2.0f * current_length - x_next;
1001           dt = 0.0f;
1002         } else {
1003           // x_next < 0.0f -- bounce
1004           if (x != 0.0f) {
1005             // first bounce
1006             bounce_instance++;
1007           }
1008           bounce_count++;
1009
1010           float a = drift * dt;
1011           float b = sigma * sqrt_dt * w;
1012           float sqrt_alpha = solve_quadratic(a, b, x);
1013           float alpha = sqrt_alpha * sqrt_alpha;
1014
1015           dt *= (1.0f - alpha);
1016           float rand_val = curand_uniform(&local_state);
1017           int new_edge = 0;
1018           while (new_edge < num_edges - 1 && rand_val > jump_weights[new_edge]) {
1019             new_edge++;
1020           }
1021
1022           if (new_edge < 0 || new_edge >= num_edges) {
1023             printf("invalid new_edge %d, clamping to 0\n", new_edge);
1024             new_edge = 0;
1025           }

             edge = new_edge;
             x = 0.0f;
         }
```

```
1026            }
1027        }
1028
1029      edges[tid] = edge;
1030      positions[tid] = x;
1031      bounces[tid] = bounce_count;
1032      bounce_instances[tid] = bounce_instance;
1033      states[tid] = local_state;
1034  }
1035
1036  __global__ void langevin_multi_step_graph_kernel(
1037      int *edges, float *positions, int *bounces, int *bounce_instances,
1038      const float *edge_lengths, const float *drift_coeffs,
1039      const int2 *edge_vertices, const int *vertex_edge_offsets,
1040      const int *vertex_edge_indices, const int *vertex_edge_orientations,
1041      const float *vertex_edge_cumweights, const int potential_type,
1042      const float base_dt, const float sigma, const int num_edges,
1043      const int num_particles, curandstate *states) {
1044    const int tid = blockidx.x * blockdim.x + threadidx.x;
1045    if (tid >= num_particles)
1046      return;
1047
1048    int edge = edges[tid];
1049    float x = positions[tid];
1050    int bounce_count = bounces[tid];
1051    int bounce_instance = bounce_instances[tid];
1052    curandstate local_state = states[tid];
1053
1054    if (edge < 0 || edge >= num_edges) {
1055      printf("invalid initial edge %d for particle %d\n", edge, tid);
1056      edge = 0;
1057    }
1058
1059    for (int step = 0; step < steps_per_kernel; ++step) {
1060      float dt = base_dt;
1061      int iterations = 0;
1062
1063      while (dt > 0.0f && iterations++ < max_iterations_per_step) {
1064        float w = curand_normal(&local_state);
1065
1066        float drift = compute_drift(potential_type, edge, x, drift_coeffs);
1067        float sqrt_dt = sqrtf(dt);
1068        float x_next = x + dt * drift + sigma * sqrt_dt * w;
1069        float current_length = edge_lengths[edge];
1070
1071        if (current_length <= 0.0f) {
1072          printf("invalid edge length %f for edge %d\n", current_length, edge);
1073          current_length = 1.0f;
1074        }
1075
1076        if (x_next > 0.0f && x_next <= current_length) {
1077          x = x_next;
1078          dt = 0.0f;
1079        } else {
1080          bool hit_start = x_next <= 0.0f;
1081          int bounce_vertex =
1082              hit_start ? edge_vertices[edge].x : edge_vertices[edge].y;
1083          if (bounce_vertex < 0) {
```

```
1080              bounce_vertex =
1081                  hit_start ? edge_vertices[edge].x : edge_vertices[edge].y;
1082          }
1083
1084          if (x != 0.0f && x != current_length) {
1085              bounce_instance++;
1086          }
1087          bounce_count++;
1088
1089          float a = drift * dt;
1090          float b = sigma * sqrt_dt * w;
1091          float sqrt_alpha =
1092              solve_quadratic(a, b, hit_start ? x : current_length - x);
1093          float alpha = sqrt_alpha * sqrt_alpha;
1094          dt *= (1.0f - alpha);
1095          int start = vertex_edge_offsets[bounce_vertex];
1096          int end = vertex_edge_offsets[bounce_vertex + 1];
1097          int degree = end - start;
1098
1099          if (degree > 0) {
1100              float rand_val = curand_uniform(&local_state);
1101              int choice = end - 1;
1102              for (int idx = start; idx < end; ++idx) {
1103                  if (rand_val <= vertex_edge_cumweights[idx]) {
1104                      choice = idx;
1105                      break;
1106                  }
1107              }
1108              int new_edge = vertex_edge_indices[choice];
1109              int orientation = vertex_edge_orientations[choice];
1110              if (new_edge < 0 || new_edge >= num_edges) {
1111                  new_edge = edge;
1112              }
1113              edge = new_edge;
1114              float new_length = edge_lengths[new_edge];
1115              x = (orientation == 0) ? 0.0f : new_length;
1116          } else {
1117              x = hit_start ? 0.0f : current_length;
1118          }
1119      }
1120  }
1121
1122  edges[tid] = edge;
1123  positions[tid] = x;
1124  bounces[tid] = bounce_count;
1125  bounce_instances[tid] = bounce_instance;
1126  states[tid] = local_state;
1127 }
1128
1129 __global__ void setup_kernel(curandstate *states, unsigned long long seed,
1130                              int num_particles) {
1131   int tid = blockidx.x * blockdim.x + threadidx.x;
1132   if (tid >= num_particles)
1133     return;
     curand_init(seed + tid, 0, 0, &states[tid]);
 }
```

```
// new histogram kernel
__global__ void compute_histogram_kernel(const int *edges,
                                         const float *positions,
                                         const float *edge_lengths,
                                         const int *bin_offsets,
                                         const float *bin_lengths,
                                         int *histograms,
                                         int num_edges, int num_particles) {
  const int tid = blockidx.x * blockdim.x + threadidx.x;
  if (tid >= num_particles)
    return;

  const int edge = edges[tid];
  const float pos = positions[tid];

  if (edge < 0 || edge >= num_edges)
    return;
  const float length = edge_lengths[edge];
  if (pos < 0.0f || pos > length)
    return;

  const int start = bin_offsets[edge];
  const int end = bin_offsets[edge + 1];
  if (start >= end)
    return;

  float accum = 0.0f;
  int bin = start;
  for (int i = start; i < end; ++i) {
    const float blen = bin_lengths[i];
    if (blen <= 0.0f)
      continue;
    accum += blen;
    bin = i;
    // assign to the first bin whose upper boundary exceeds the position.
    if (pos <= accum)
      break;
  }

  atomicadd(&histograms[bin], 1);
}

// histogram when each edge has uniform bin width.
__global__ void compute_histogram_uniform_kernel(const int *edges,
                                                 const float *positions,
                                                 const int *bin_offsets,
                                                 const int *bin_counts,
                                                 const float *bin_widths,
                                                 int *histograms,
                                                 int num_edges,
                                                 int num_particles) {
  const int tid = blockidx.x * blockdim.x + threadidx.x;
  if (tid >= num_particles)
    return;

  const int edge = edges[tid];
  if (edge < 0 || edge >= num_edges)
```

```
1188        return;
1189
1190    const int count = bin_counts[edge];
1191    if (count <= 0)
1192        return;
1193
1194    const float width = bin_widths[edge];
1195    if (width <= 0.0f)
1196        return;
1197
1198    const float pos = positions[tid];
1199    int bin_local = (int)(pos / width);
1200    if (bin_local >= count)
1201        bin_local = count - 1;
1202    if (bin_local < 0)
1203        bin_local = 0;
1204
1205    const int bin = bin_offsets[edge] + bin_local;
1206    atomicadd(&histograms[bin], 1);
1207    }
        }
```