# OpenReview forum: "Sampling On Metric Graphs"
_ICLR.cc/2026/Conference — Submitted to ICLR 2026_

### Official Review · Reviewer_uQfq · 2025-10-27

**Soundness:** 3
**Presentation:** 3
**Contribution:** 1
**Rating:** 2
**Confidence:** 4

**Summary:**

The paper touches on a very interesting problem with solid research dealing with differential equations posed on metric graphs. Nevertheless, I feel that ICLR by its core definition is the wrong venue for this research and it is better suited for numerical analysis journal/UQ journal.

**Strengths:**

I think the paper is well written and deals with an interesting problem but I do not see it as being particularly focused on learning algorithms and doubt its suitability for ICLR.

**Weaknesses:**

I feel that the poor performance of the FVM needs to be discussed and these methods usually perform very well on metric graphs but have, of course, other weaknesses. The formatting of all the references is terrible please fix the bib-entries.

**Questions:**

- Please explain how changing the orientation of an edge does not change the inward derivative?
- In (1) and (2) the subscript e is not explained. It is clearly the restriction onto the edge but please be precise.
- I am a bit suspicious about the FVM scheme as it seems to completely fail for the Fokker Planck equation, any reasons why? There are existing FVM schemes for quantum graphs to be found on Github that seem to do well for other PDEs.

---

> ### Author Response · Authors · 2025-11-30
> **Response to reviewer uQfq**
>
> We thank the reviewer for their useful feedback and truly appreciate the time taken by them to evaluate our paper. We respond to their concerns below.
>
> *Inward derivative and edge orientation clarification*
>
> We refer the reviewer to figure 2 for help in understanding inward derivatives. Consider edge $e_1$. If its orientation is such that the vertex $v$ is the initial vertex, then the inward derivative is defined as $-\frac{\partial f_e}{\partial x} (0)$. Note that $f_e$ here has increasing $x$ away from the vertex, and hence we have a minus sign to flip the derivative, since we want the inward derivative to represent the change in $f$ as we move $x$ towards the vertex. On the other hand, if the orientation is flipped and $v$ is the terminal vertex, then the inward derivative is $\frac{\partial f_e}{\partial x} (l_e)$. Note here that the derivative is evaluated at the same point as before (since $x = l_e$ now corresponds to $v$ instead of $x=0$). However, the direction of increasing $x$ is now *towards* the vertex $v$, so the derivative is a different sign. But in our definition of inward derivative, we no longer have a minus sign, so we maintain the same sign and capture the rate of change of $f$ as $x$ moves towards the vertex again (which is the inward derivative). We hope this helps.
>
> *Extension to General graphs*
>
> We added Section 3.2 in the main paper, as well as Algorithm 2 in the appendix which details how to extend our algorithm to general graphs. We have now also provided CUDA implementations of this general graph algorithm (source code in the Appendix as well as revised supplementary material). Further, we added Section 4.2, where we benchmark our algorithm on a real cortical vascular network with ~1.5k nodes and edges with cycles and heterogeneous lengths, obtained from the DuMuX repository of examples. We simulate advection--diffusion of tracer transport on this network using our algorithm. We compare to a strong software baseline, which is the DuMuX FVM, and still achieve ~1500x speedups while also being able to handle larger timestep sizes beyond the stability limit of the FVM solver. Figure 5 shows the runtime scaling against the DuMuX FVM baseline, as well as the computed steady state density on the network layout. Further details are provided in the revised paper.
>
> *FVM Baseline*
>
> As part of the extension to general graphs, we used a strong thorough FVM baseline that uses an explicit scheme with Newton iterations to solve each timestep. We demonstrate on a real metric graph that this method inevitably fails to physical stability limits as we increase the timestep size, or conversely, increase the resolution of the mesh (reduce the finite volume cell size). Our algorithm does not suffer any such stability issue at finer resolution. However, we would like to be clear that this scheme does not completely fail, but achieves a very reasonable and fast baseline when within stability limits, but fails to utilize the GPU efficiently which our algorithm leverages strongly. We hope this clarifies the reviewer's concerns.
>
> *Suitability for ICLR*
>
> We would like to highlight that numerous applications in the practical sciences like quantum physics, biology and neuroscience are well modeled my metric graphs.
> Our main algorithm introduces the first method for sampling on these domains. This serves as a stepping stone for developing both machine learning and probabilistic methods, and future work can connect to stochastic optimization as well. Additionally, our algorithm has theoretical guarantees for both consistency and practical feasibility.
> Finally, we also present a fast, memory-aware CUDA implementation of our algorithm and demonstrate that it is practical, scalable, and amenable to modern computing infrastructure.
> We believe that these aspects are well suited to the nature of work published in ICLR and thus make our paper a good fit for this venue.

---

### Official Review · Reviewer_E8PR · 2025-11-01

**Soundness:** 2
**Presentation:** 2
**Contribution:** 3
**Rating:** 8
**Confidence:** 2

**Summary:**

This paper tackles sampling on metric graphs by introducing a time-step–splitting Euler–Maruyama (EM) scheme to simulate Brownian motion, and hence Langevin diffusions, on metric graphs. It proves that the number of vertex crossings is finite with high probability and that exit probabilities of the simulation converge to the SDE's vertex-edge jump probabilities as the step size tends to zero. The authors also present highly parallel, memory-aware implementations, and experiments show the method outperforms a finite-volume baseline in accuracy and speed.

**Strengths:**

1. The paper introduces a novel and interesting time-step–splitting EM scheme for Brownian/Langevin on metric graphs, with guarantees of finite splits (w.h.p.) and exit-probability convergence as step size goes to 0.

2. This method is efficient and scalable with memory-aware, highly parallel GPU implementation that showing strong speedups and accuracy gains over a finite-volume baseline with empirical validation

**Weaknesses:**

Results are interesting but restricted to star graphs and applicability to general metric graphs is neither analyzed nor empirically validated.

Beyond finite splits and exit-probability consistency, there are no non-asymptotic weak/strong error bounds or sampling error rates.

Minors:

- "Sampling On Metric Graphs" should be "Sampling on Metric Graphs"

- Line 405 the normalizing constant B if given by -> the normalizing constant B is given by

**Questions:**

Q1: What's the non-asymptotic convergence rate?

Q2: What are the challenges for non-star graphs, and can you show experiments on non-star graphs?

---

> ### Author Response · Authors · 2025-11-30
> **Response to reviewer E8PR**
>
> We thank the reviewer for their useful feedback and truly appreciate the time taken by them to evaluate our paper. We respond to their concerns below.
>
> *Non-asymptotic convergence.*
>
> On each edge, the discretization reduces to Euler--Maruyama and inherits strong order $1/2$ convergence under locally Lipschitz drift/diffusion.
> A full graph-wide non-asymptotic sampling rate would require bounding bounce-induced path variation and is significantly more challenging and sophisticated -- this remains open and is a future line of work we are actively looking into.
>
> *Extension to General Graphs*
>
> We added Section 3.2 in the main paper, as well as Algorithm 2 in the appendix which details how to extend our algorithm to general graphs. We have now also provided CUDA implementations of this general graph algorithm (source code in the Appendix as well as revised supplementary material). Further, we added Section 4.2, where we benchmark our algorithm on a real cortical vascular network with ~1.5k nodes and edges with cycles and heterogeneous lengths, obtained from the DuMuX repository of examples. We simulate advection--diffusion of tracer transport on this network using our algorithm. We compare to a strong software baseline, which is the DuMuX FVM, and still achieve ~1500x speedups while also being able to handle larger timestep sizes beyond the stability limit of the FVM solver. Figure 5 shows the runtime scaling against the DuMuX FVM baseline, as well as the computed steady state density on the network layout. Further details are provided in the revised paper.
>
> We have fixed the minor typos in the revision as well, thank you for catching them.

---

### Official Review · Reviewer_se1y · 2025-11-01

**Soundness:** 3
**Presentation:** 2
**Contribution:** 2
**Rating:** 4
**Confidence:** 3

**Summary:**

This paper investigated the practical implementation of sampling in metric graphs. It seems that the authors generalise the Euler-Maruyama discretization of Langevin diffusion for continuous distribution. They provide a theoretical guarantee that the jump distribution generated by their proposed algorithm asymptotically converges to the target distribution. Then they provide a parallelizable version of the proposed algorithm to get fast and memory-saving implementation.

**Strengths:**

1. The problem investigated in this paper seems to be novel, having theoretical and practical value.

2. This paper has a certain mathematical depth.

**Weaknesses:**

1. There are some claims that they did not explain clearly. For example, in Line 133-134, they claimed that the results of the star graphs researched in this paper can extend to general graphs. They did not explain how to extend.

2. The presentation should be improved. The key section “Brownian Motion on Metric Graphs” should be more detailed, especially on the boundary conditions, which is not friendly to the readers.

**Questions:**

The theoretical results seems to be valid for all target distribution $b_v$, which means they did not need to make any assumption on $b_v$. This is different from Unadjusted Langevin Algorithm for sampling in continuous distribution, which usually make assumptions on smoothness and isoperimetry properties of potential function.

---

> ### Author Response · Authors · 2025-11-30
> **Response to reviewer se1y**
>
> We thank the reviewer for their useful feedback and truly appreciate the time taken by them to evaluate our paper. We agree with the reviewer's points about empirical evaluation and general graph extension and have now included a comprehensive evaluation in the revision.
>
> *Extension to General Graphs*
>
> We added Section 3.2 in the main paper, as well as Algorithm 2 in the appendix which details how to extend our algorithm to general graphs. We have now also provided CUDA implementations of this general graph algorithm (source code in the Appendix as well as revised supplementary material).
> Further, we added Section 4.2, where we benchmark our algorithm on a real cortical vascular network with ~1.5k nodes and edges with cycles and heterogeneous lengths, obtained from the DuMuX repository of examples. We simulate advection--diffusion of tracer transport on this network using our algorithm. We compare to a strong software baseline, which is the DuMuX FVM, and still achieve ~1500x speedups while also being able to handle larger timestep sizes beyond the stability limit of the FVM solver.
> Figure 5 shows the runtime scaling against the DuMuX FVM baseline, as well as the computed steady state density on the network layout. Further details are provided in the revised paper.
>
> *Exposition of Brownian Motion*
>
> We have updated some of the phrasing and flow of the exposition section that would now hopefully make this section more approachable.
>
> *Clarification on $b_v$*
>
> $b_v$ here is more a part of the underlying graph structure rather than the fields and densities on the graph. For example, in our experiment on the cortical network, it represents the ratios of cross-sectional areas of the blood vessels incident at a vertex. This effectively re-weights how a diffusing particle would move when starting at the vertex (it is more likely to jump to a vessel with a larger cross section than a smaller one). As such, this does not correspond to drift potentials on the graph. In fact, for our theoreticals, we make similar regularity assumptions as ULA on the smoothness of the potentials on the graph (see the inputs of Algorithm 1 for instance).

---

### Official Review · Reviewer_Wjmx · 2025-11-01

**Soundness:** 3
**Presentation:** 2
**Contribution:** 2
**Rating:** 4
**Confidence:** 2

**Summary:**

This paper focuses on the numerical simulation problem of Brownian motion and Langevin diffusions on metric graphs and proposes a timestep splitting Euler-Maruyama-based discretization method.
It handles vertex crossings by recursively splitting the simulation step, moving the particle to the vertex, and sampling a new edge. Theoretical analysis prove the algorithm terminates finitely and converges to correct vertex-edge jump probabilities as the timestep decreases.
A custom, memory-aware CUDA kernel is implemented for fast, parallelized execution on GPUs.
Numerical experiments on a 5-edge star graph with linear and quadratic potentials demonstrated the effectiveness of the proposed method.
The reported results show significant speedups and higher accuracy in recovering steady-state densities compared to FVM.

**Strengths:**

1. Though the theory of function space and Brownian motion on metric graphs is well-established, practical simulation algorithm is non-existent. This paper provides a concrete and practical method to simulate this process, which is a novel contribution to the field and will be beneficial for future work in this field.

2. Theoretical analysis is thorough and insightful. Theorem 2 & 3 addresses the concern of an infinite loop due to repeated vertex crossings within a single timestep, which is crucial for the practicality of the algorithm.  Corollary 1 links the algorithm's behavior to the underlying SDE, proving that the simulated jump probabilities converge to the correct theoretical values $b_v$. These theoretical analysis provide a solid ground for the proposed method.

3. The CUDA kernel implementation achieves a massive speedup over a simple Pytorch implementation. This engineering effort significantly elevates the paper's utility for practitioners and researchers needing large-scale simulations.

**Weaknesses:**

1. Experimental Evaluation is limited critically. The entire numerical evaluation is conducted on a synthetic star graph with only 5 edges. Metric graphs are powerful precisely for modeling networks with complex cycles, multiple vertices, and varied edge lengths. Demonstrating performance only on a star graph provides almost no evidence that the algorithm works on metric graphs in general. Besides, this paper does not demonstrate the effectiveness of algorithm on a real-world problem or dataset, which undermines its potential impact. The performance gap versus the FVM baseline, while impressive, is less meaningful without a real-world context.

2. While Theorem 2 guarantees finite runtime, empirical analysis of the computation cost introduced by vertex crossings is missed. How does the average number of splits M scale with $\delta t$, the drift magnitude, and the graph complexity? This is an important practical consideration that is left unexplored.

**Questions:**

1. Can you provide theoretical or experimental analysis to show that Algorithm 1 works effectively on a non-star metric graph, for instance, a graph containing a cycle or multiple interconnected vertices?

2. Can you provide experimental results on a concrete real-world dataset?

3. This paper focuses exclusively on standard boundary conditions, which can be further improved by discussing limitations of the current algorithm regarding these more general conditions or outlined a path for future extension.

---

> ### Author Response · Authors · 2025-11-30
> **Response to reviewer Wjmx**
>
> We thank the reviewer for their useful feedback and truly appreciate the time taken by them to evaluate our paper. We agree with the reviewer's points about empirical evaluation and have now included a comprehensive evaluation in the revision.
>
> *Real non-star experiment.*
>
> We added Section 4.2, where we benchmark our algorithm on a real cortical vascular network with ~1.5k nodes and edges with cycles and heterogeneous lengths, obtained from the DuMuX repository of examples. We simulate advection--diffusion of tracer transport on this network using our algorithm. We compare to a strong software baseline, which is the DuMuX FVM, and still achieve ~1500x speedups while also being able to handle larger timestep sizes beyond the stability limit of the FVM solver.
> Figure 5 shows the runtime scaling against the DuMuX FVM baseline, as well as the computed steady state density on the network layout. Further details are provided in the revised paper.
> This involved extending our algorithm to the general graph case (which is Algorithm 2 in the Appendix), and we have implemented the corresponding CUDA kernels as well. You can find the updated source code both in the Appendix and in the revised supplementary material.
>
>
>
> *Boundary conditions / general graphs.*
>
> We focus on standard gluing/flux balance. The revised appendix provides details on the general-graph extension.
> Other vertex conditions would change the nature of the stochastic process, allowing particles to spend time at vertices and then we would need to model a waiting-time distribution for the particle to escape the vertex. We decided against studying this setting both for clarity and focus of presentation, and also because this setting is less representative of real world metric graphs.
>
> *Distribution of $M$*
>
> Following the analysis in Appendix A.2.1, we have a lower bound on the CDF of $M$ -- specifically that it is lower bounded by the tail of a $\chi^2$ distribution. As a consequence, we can observe that the distribution of bounces is sub-exponential. The complexity and properties of the graph is captured in the quantity $\gamma$, which effectively captures the "worst-case" impact of the drift, diffusion, and timestep size over the whole graph.

---

### Meta-Review · Area_Chair_x3SD · 2025-12-27

**Summary:**

The authors propose a timestep splitting Euler–Maruyama discretization scheme for simulating Brownian motion and Langevin diffusions on metric graphs, with theoretical guarantees on the number of timestep splittings and convergence of the exit probabilities to the vertex-edge jump probabilities. Additionally, the authors provide a highly optimized CUDA implementation that achieves large speedups over finite-volume and PyTorch-based baselines. In the rebuttal, the authors extend the method to general metric graphs (beyond the start metric graphs), and additional experimental results on a real cortical vascular network.

The Reviewers think that the theoretical finding results are interesting; the provided CUDA implementation will be helpful for subsequent research. However, the current presentation heavily focuses on the special star metric graph, and certain extension provided in the rebuttal. We think that it is better to frame the considered problem and experimental setting with general metric graph.

In brief, although the proposed ideas and theoretical finding results are appreciated by the Reviewers; its empirical support evidence and the presentation limit its contribution, which falls short to the bar. We think it is necessary for one more round of review. The authors may consider the Reviewers' comments to improve the work.

**Reviewer Concerns:**

The Reviewers have some following concerns:

+ Reviewer Wjmx: weak experimental settings; lack empirical evidence on real-world tasks; weak empirical evidence for theoretical finding results (Theorem 2); general setting beyond standard boundary conditions;

+ Reviewer se1y: clarification for general metric graph extension, beyond the focused star metric graph; presentation on the main technical finding results.

+ Reviewer E8PR: restriction on star metric graph; extension to general metric graphs; lacking error bounds / sampling error rates beyond finite splits and exit-probability consistency.

+ Reviewer uQfq: ICLR scope; presentation format; empirical analysis on strong/weak points; performances of FVM

**Reviewer Scores:**

The authors provide additional empirical results on non-start metric graph; provides certain extension to general metric graph, beyond the considered star metric graph; clarification on the standard boundary conditions for problem settings.

We think the additional empirical results on non-star metric graphs and certain extension of the results to general metric graph improves the submissions. However, it is better to frame the considered problem to general metric graph instead of the special star metric graph; and describe these extensions rigorously as the main contributions.

---

### Decision · Program_Chairs · 2026-01-26

Reject